# $\mathtt{bp}$: Blood pressure analysis in R

**John Schwenck****[1¤*], Naresh M. Punjabi[2], Irina Gaynanova[1]**

**1** Department of Statistics, Texas A&M University, College Station, TX, United States of America, **2** Miller School of Medicine, University of Miami, Miami, FL, United States of America

¤ Current address: AI/ML Center of Excellence, Wells Fargo, Charlotte, NC, United States of America
* jschwenck12@gmail.com

## Abstract

Despite the world-wide prevalence of hypertension, there is a lack in open-source software for analyzing blood pressure data. The R package $\mathtt{bp}$ fills this gap by providing functionality for blood pressure data processing, visualization, and feature extraction. In addition to the comprehensive functionality, the package includes six sample data sets covering continuous arterial pressure data (AP), home blood pressure monitoring data (HBPM) and ambulatory blood pressure monitoring data (ABPM), making it easier for researchers to get started. The R package $\mathtt{bp}$ is publicly available on CRAN and at https://github.com/johnschwenck/bp.

**Data Availability Statement:** The data underlying the results presented in the study are available as part of the bp package. The package can be accessed from https://CRAN.R-project.org/package=bp and from https://github.com/johnschwenck/bp.

## Introduction

Cardiovascular disease remains the leading cause of death worldwide, with hypertension alone affecting over 1.1 billion people annually according to the American Heart Association [1]. Hypertension is characterized by elevated levels of blood pressure (BP), therefore diagnosis and effective management of hypertension relies on monitoring an individual's BP readings. On one hand, BP can exhibit pronounced changes over a long-term time horizon (weeks, months, years, etc.) due to aging or behavioral factors (e.g., smoking). On the other hand, BP exhibits noticeable fluctuations over a 24 hour period (day-night changes, sleep-awake changes, etc.), with high 24 hour BP variability being an independent predictor of cardiovascular events such as heart attack, stroke, or another potentially fatal infarction [2]. Thus, characterization of blood pressure variability (BPV), e.g. fluctuations in BP readings, depends on the chosen time-scale: *very short-term* (continuous beat-to-beat arterial BP), *short-term* (within 24 hours, typically in intervals ranging from 5 minutes to every hour), *middle-term* (day-to-day), and *long-term* (visit-to-visit, etc) [3].

Different BP measuring methods are used depending on the chosen time-scale. Continuous arterial pressure (AP) waveform data is collected at a very high sampling rate (i.e. 100 Hz) and used to assess very short-term variability. By contrast, devices that acquire data from a sphygmomanometer such as office monitors, home blood pressure monitors (HBPM), or ambulatory blood pressure monitors (ABPM) discretize the AP waveform data into summarized output. Home blood pressure monitoring (HBPM) devices or office readings are effective at measuring variability over the middle- and long-term horizons since patients are typically more willing to take measurements at a lower frequency (once or twice per day) for a longer duration (weeks, months, years). However, HBPM data are limited to measurements taken at convenient times, as the patient must initiate the recording. The resulting readings are

**Funding:** This work was supported by the National Science Foundation [NSF CAREER DMS-2044823 to I.G.]. The funder had no role in study design, data collection and analysis, decision to publish, or preparation of the manuscript.

**Competing interests:** The authors have declared that no competing interests exist.

unequally spaced in time, as they lack the consistent frequency of automatic measuring devices. Ambulatory blood pressure monitoring (ABPM) devices are used for short-term measurements as they take automatic readings from patient in a free environment at the pre-specified intervals (e.g. 30 min) over a 24-hour period or longer. A distinct advantage of ABPM compared to other methods is that they can provide nocturnal measurements at night in addition to diurnal measurements during the day.

The complexity and variety of BP measurements data pose computational challenges for clinicians and researchers. Substantial effort is required to identify the proper BP data format, visualize the data, and calculate various metrics including the overall average, the variance, and temporal differences. Existing software for analyses of temporal BP data has several limitations. First, most of the available software is proprietary (e.g. AccuWin Pro™ by SunTech Medical, CardioSoft™ by GE Healthcare) with associated costs. Second, the "black-box" nature of underlying calculations makes these software undesirable for researchers looking for accessible and reproducible solutions. Third, the existing mobile applications designed for the patient's self-management (e.g. OMRON connect, Withings Health Mate, QARDIO) lack export capabilities and are unable to calculate BP metrics beyond standard summary statistics for customized analysis. Finally, the existing open source software for BP data, e.g. Python package pyPDA [4], have limited functionality (e.g. lack visualization capabilities, small set of BP variability measures). Thus, there remains a need for a comprehensive open-source software for the analysis of BP data.

The objective of this work is to introduce bp—an open-source (GPL-3) R package dedicated to analyzing blood pressure data, and the first R package designed for this purpose. We chose to use R software platform over existing alternatives due to several reasons. First, R is a script-based programming language, which allows the creation of reproducible scripts for all BP data processing and metric calculation steps. This is not possible with point and click software solutions, such as Excel or SPSS statistical software. Second, while SAS and SPSS statistical software are commonly used for analyses of blood pressure datasets, R is also gaining popularity as evidenced by its use in multiple studies involving blood pressure data [5–9]. This trend in using R for BP data sets is consistent with R being one of the most popular and heavily utilized data analysis platforms [10, 11], with [12] providing detailed guidelines and comparison of functionality for SAS and SPSS software users who wish to use R. Third, being free and open-source, R has an incredibly rich set of data analysis and graphics tools, with over 5,000 add-on packages available in addition to its base functionality. We are able to take advantage of this functionality in designing customized functions and visualizations. Specifically, the bp package has built-in automatic processing functionality for three main types of blood pressure data: arterial pressure (AP) continuous waveform recordings, home blood pressure monitoring (HBPM) and ambulatory blood pressure (ABPM). The bp package also allows one to assess the dynamics of blood pressure changes by implementing quantitative BP metrics such as ARV [13], wSD [14], morning BP surge [15], dipping [16] and several others. Finally, bp has multiple visualization capabilities to assist with data interpretation.

In this paper, we summarize the BP data sets included with the package, describe the package's automatic data processing functionality, and provide an overview of implemented BP metrics and visualizations. To illustrate the package functionality in practice, we highlight two case studies that analyze both a single-subject data set with HBPM measurements (Case Study I), and a multi-subject data set with ABPM measurements (Case Study II).

## Sample data sets

The bp package includes several BP data sets as examples of commonly used data formats. The data sets available as of bp version 2.0.1 are listed in Table 1.

**Table 1. Summary of data sets included in the bp package; AP refers to arterial (Blood) pressure, HBPM refers to home blood pressure monitoring, and ABPM refers to ambulatory blood pressure monitoring.**

| Name | Type | Format | Description |
|------|------|--------|-------------|
| bp_rats | AP | Multi-Subject | *Hypertension in salt-sensitive Dahl rats* [17] |
| bp_children | HBPM | Multi-Subject | *B-Proact1v Child BP Study* [18] |
| bp_ghana | HBPM | Multi-Subject | *Effects of health insurance coverage and task-shifting strategy on patients with uncontrolled hypertension in Ghana* [19] |
| bp_jhs | HBPM | Single-Subject | *Blood pressure during endured aerobic exercise (NYC to AK cycling)* [20] |
| bp_preg | HBPM | Multi-Subject | *Pregnancy-induced hypertension and pre-eclampsia prediction* [21] |
| bp_hypnos | ABPM | Multi-Subject | *ABPM profiles of 5 subjects with type 2 diabetes and obstructive sleep apnea* [22] |

bp_rats: The data set contains AP measurements sampled at 100 Hz of the SS ($n_1 = 9$) and SS.13 ($n_2 = 6$) genetic strains of Dahl rats. Each mouse was administered either a low sodium or a high sodium diet. The aim of the study [17] was to investigate the connection between the dysfunction of the baroreflex control system in Dahl rats and salt-sensitive hypertension.

bp_children: The data set contains HBPM measurements on 1, 283 children from Bristol, UK. Three blood pressure readings per visit were collected (at 9 years of age and 11 years of age). Additionally, information on their physical activity has been collected. The aim of the study [18] was to examine how sedentary behavior influences children progressing through primary school, and to understand the association between elevated blood pressure in children and its impact on the development of cardiovascular disease into adulthood.

bp_ghana: The data set contains HBPM measurements on 757 subjects across 32 community health centers from a cluster-randomized trial in Ghana: 389 subjects were in the health insurance coverage (HIC) group and 368 subjects were in another group consisting of a combination of HIC with a nurse-led task-shifting strategy for hypertension control (TASSH) (this group is denoted TASSH + HIC). Baseline blood pressure measurements were collected, with 85% of subjects having 12 month follow-up measurements available. The aim of the study [19] was to assess the comparative effectiveness of HIC alone versus the combination of TASSH + HIC on reducing systolic blood pressure among patients with uncontrolled hypertension.

bp_jhs: The data set consists of HBPM measurements from a single subject who endured aerobic (endurance) exercise over 95 days by cycling 5,775 miles from New York City to Seward, Alaska. Data was collected using an Omron Evolv wireless blood pressure monitor twice daily (in the morning upon waking up and in the evening before bed). The aim of the study [20] was to assess blood pressure variability.

bp_preg: The data set contains HBPM measurements from 209 women, each of whom were assessed every 30 minutes during the Pregnancy Day Assessment Clinic (PDAC) observation window for up to 240 minutes (i.e. up to 8 total readings per subject per observation window) in addition to an initial reading prior to the assessment. The aim of the study [21] was to investigate the pregnancy-induced hypertension and pre-eclampsia prediction to determine whether the blood pressure assessment of the first 1 hour (60 minutes) observation window is sufficiently accurate relative to the standard 4 hour (240 minute) window.

bp_hypnos: The data set contains ABPM measurements from 5 subjects with type 2 diabetes and obstructive sleep apnea. The recordings are taken every hour during the 24-hour period for each of the two visits 3 months apart. Additional information includes sleep/wake indicator for each recording inferred from a wrist-worn actigraphy device. The aim of the study [22] was to determine the effect of positive airway pressure treatment of sleep apnea on glycemic control and blood pressure of patients with type 2 diabetes. The included data is a subset of the full data described in [22].

## Data processing

All of the functions in the bp package assume that the data has been processed in a common format. To reduce the required data processing burden, the package includes a process_data function that transforms the input data source into a data frame with a unified format. The underlying functionality includes standardization of the column names, formats for dates and times, and classification of readings into blood pressure stages among other processing steps. The process_data function should be applied first before performing any other analyses using the bp package.

As an example, consider the bp_jhs and bp_hypnos data sets. These data sets correspond to different types of BP data (HBPM and ABPM, respectively), have different column names, and not all of the same variables are present in both data sets.

**Code Block 1**

```
> # Check column names of both data sets:

> names(bp_jhs)
 [1] "DateTime" "Month" "Day" "Year" "DayofWk" "Hour" "Meal_Time"
 [8] "Sys.mmHg." "Dias.mmHg." "bpDelta" "Pulse.bpm."

> names(bp_hypnos)
 [1] "NR." "DATE.TIME" "SYST" "MAP" "DIAST" "HR" "PP" "RPP" "WAKE" "ID"
[11] "VISIT" "DATE"
```

These inconsistencies are eliminated after applying process_data function:

**Code Block 2**

```
> # Create an object jhs_proc to contain the processed bp_jhs data. This
> # object adheres to the standardization protocols of the process_data
> # function to be used in future analyses

> jhs_proc <- process_data(bp_jhs, bp_type = "hbpm",
                                        sbp = "Sys.mmHg.",
                                        dbp = "Dias.mmHg.",
                                  date_time = "DateTime",
                                         hr = "Pulse.bpm.")

> # Repeat for bp_hypnos data:

> hypnos_proc <- process_data(bp_hypnos, bp_type = "abpm",
                                               sbp = "SYST",
                                               dbp = "DIAST",
                                         date_time = "DATE.TIME",
                                                hr = "HR",
                                                pp = "PP",
                                               map = "MAP",
                                               rpp = "RPP",
                                                id = "ID",
                                             visit = "VISIT",
                                              wake = "WAKE")
```

First, the user must specify the type of blood pressure data ("ap", "hbpm" or "abpm") via the bp_type argument, where the default is "hbpm". Specifying the correct type is crucial as

the behavior of some package functionality changes depending on the type. For example, nocturnal metrics, such as dip_calc or mbps (refer to *Nocturnal BP metrics* section), can be used with ABPM data (which contain nocturnal recordings) but not with HBPM data (which cannot contain nocturnal readings as the user must be awake to initiate the recording). The time of nocturnal readings is determined from either the user-supplied WAKE column (based on additional information from an actigraphy device), or the ToD_int parameter (based on time of day) in the absence of WAKE. The default nocturnal period (when WAKE = 0) ranges from 12AM to 6AM.

Second, the user must specify the column names corresponding to sbp (systolic blood pressure) and dbp (diastolic blood pressure). The specification of the remaining columns names (i.e. heart rate hr) is optional based on the data available in a specific dataset. It is recommended that the user specifies as many columns as possible.

The returned processed data frames have standardized and capitalized column names together with additional variables created as part of the processing.

**Code Block 3**

```
> names(jhs_proc)
 [1] "ID"            "GROUP"        "DATE_TIME"    "DATE"
 [5] "DAY_OF_WEEK"   "YEAR"         "MONTH"        "DAY"
 [9] "HOUR"          "TIME_OF_DAY"  "SBP"          "DBP"
[13] "BP_CLASS"      "HR"           "MAP"          "RPP"
[17] "PP"            "DAYOFWK"      "MEAL_TIME"    "BPDELTA"
[21] "SBP_CATEGORY"  "DBP_CATEGORY" "bp_type"

> # Notice the added variables in the hypnos_proc variable below
> # that are automatically derived

> names(hypnos_proc)
 [1] "ID"            "GROUP"        "DATE_TIME"    "DATE"
 [5] "DAY_OF_WEEK"   "YEAR"         "MONTH"        "DAY"
 [9] "HOUR"          "TIME_OF_DAY"  "SBP"          "DBP"
[13] "BP_CLASS"      "HR"           "MAP"          "MAP_OLD"
[17] "RPP"           "PP"           "PP_OLD"       "WAKE"
[21] "VISIT"         "NR."          "SBP_CATEGORY" "DBP_CATEGORY"
[25] "bp_type"
```

The Rate Pressure Product (RPP), Pulse Pressure (PP), and Mean Arterial Pressure (MAP) columns all did not previously exist in the original bp_jhs data set, yet were automatically created by process_data as RPP = SBP * HR, PP = SBP − DBP and MAP = (SBP + 2×DBP)/3. Furthermore, the date and time information is parsed to create separate columns for DATE, MONTH, DAY, YEAR, HOUR, DAY_OF_WEEK and TIME_OF_DAY which are used for generating summary statistics (refer to *Metrics and visualizations* section). Here, TIME_OF_DAY is one of "Morning", "Afternoon", "Evening" or "Night".

## Classification of blood pressure into stages

In clinical practice, it is common to characterize blood pressure into stages based on the BP readings. The explicit characterization of the stages, however, may change with time as new findings emerge.

Recent guidelines from the American Heart Association [23] suggest five stages: Normal (SBP < 120 and DBP < 80), Elevated / Pre-Hypertension (SBP ≥ 120 and DBP < 80), Stage 1

**Table 2. BP stages from [9] with (optional) low and crisis categories.** ISH refers to Isolated Systolic Hypertension, IDH refers to Isolated Diastolic Hypertension.

| BP Stage | Systolic (mmHg) | | Diastolic (mmHg) |
|---|---|---|---|
| Low (optional) | < 100 | and | < 60 |
| Normal | < 120 | and | < 80 |
| Elevated | 120–129 | and | < 80 |
| Stage 1—All | 130–139 | and | 80–89 |
| Stage 1—ISH (ISH—S1) | 130–139 | and | < 80 |
| Stage 1—IDH (IDH—S1) | < 130 | and | 80–89 |
| Stage 2—All | ≥ 140 | and | ≥ 90 |
| Stage 2—ISH (ISH—S2) | ≥ 140 | and | < 90 |
| Stage 2—IDH (IDH—S2) | < 140 | and | ≥ 90 |
| Crisis (optional) | ≥ 180 | or | ≥ 120 |

Hypertension (SBP $\in$ [130, 139] or DBP $\in$ [80, 89]), Stage 2 Hypertension (SBP $\geq$ 140 or DBP $\geq$ 90), and Hypertensive Crisis (SBP $\geq$ 180 or DBP $\geq$ 120). While these thresholds offer insight into an individual's hypertension severity, they do not characterize the entire spectrum of blood pressure variability. It is unclear how to classify subjects who exhibit isolated systolic or diastolic hypertensive episodes, where either the systolic value is unusually high and the diastolic value is either normal or unusually low, or vice versa. Lee et al. [9] outlines eight blood pressure stages that incorporate isolated systolic or diastolic hypertensive episode thresholds, leading to an unambiguous 2-to-1 mapping of (SBP, DBP) values to a stage as described in Table 2.

Following these guidelines, process_data function generates three columns with blood pressure classification: SBP_CATEGORY, DBP_CATEGORY and BP_CLASS. The first two separately classify SBP or DBP recording (as either "Normal", "Elevated", "Stage 1" or "Stage 2"), whereas BP_CLASS allocates the stage based on the paired value (SBP, DBP) using the classification as in [9].

The threshold values in Table 2 are based on office BP values. These thresholds are often adjusted when monitoring hypertension via out-of-office HBPM [24] and ABPM measurements [25], however there is some variability across studies and guidelines regarding the recommended adjustments. We refer the reader to Tables 3 and 4 in [26] for summary of thresholds correspondence with ABPM and HBPM, respectively. process_data function allows customization of default threshold values to match the adjustment chosen by the user via optional bp_cutoffs argument. process_data function also allows for automatic thresholds adjustment based on American Heart Association guidelines [26] by setting guidelines = "AHA".

## Metrics and visualizations

The functionality of the **bp** package revolves around recommendations outlined in O'Brien and Atkins [27] to facilitate the wider use of BP data in clinical practice. Specifically, it focuses on ease of use, visualizations, the ability to create a user-friendly one-page report, and the ability to export the output for research purposes. All implemented BP metrics and visualizations are thus available as both standalone functions and as part of an aggregated report suitable for clinical use. Table 3 provides an overview of the available metrics and visualizations in version 2.0.1 of the bp package. Note that each metric is computed separately for SBP (systolic) and DBP (diastolic) BP readings.

**Table 3. Summary of blood pressure metrics and visualizations available in the bp package.**

**Blood Pressure Metrics and Visualizations Included in bp Package**

| Function | Type | Name | Description |
|---|---|---|---|
| bp_arv | Metric | Average Real Variability | Sum of absolute differences in successive observations [13] |
| bp_cv | Metric | Coefficient of Variation | Sample mean divided by the standard deviation [28] |
| bp_sv | Metric | Successive Variation | Sum of squared differences in successive observations [28] |
| bp_center | Metric | Measures of Center | Mean and Median [29] |
| bp_mag | Metric | Magnitude | Differences between mean BP and max BP/min BP [28] |
| bp_range | Metric | Range | Max BP, min BP and range = max BP—min BP [30] |
| bp_stats | Metric | Aggregated Statistics | Combine output from arv, cv, sv, bp_center, bp_mag, and bp_range into one table |
| bp_tables | Metric | Exploratory Tables | Counts and percentages of BP readings by time of day, day of week, BP stages, etc |
| bp_sleep_metrics | Metric | Nocturnal Metrics | A list of four tables corresponding to SBP/DBP sleep period counts, averages, and nocturnal metrics [14–16, 31]:<br>• dip_calc: Nocturnal Dipping Percentage<br>• noct_fall: Nocturnal Fall<br>• wSD: Weighted Standard Deviation<br>• PW_mbps: Pre-wake MBPS<br>• ST_mbps: Sleep-Trough MBPS<br>• ME_avg: ME Average<br>• ME_diff: ME Difference |
| dip_calc | Metric | Dipping Calculation | Percent decrease in BP while asleep compared with awake along with the respective dipping classification [16] |
| dip_class_plot | Visual | Dipping Category Plot | Plot of dipping percentages by category [16] |
| bp_hist | Visual | Histograms | Histograms of BP counts and frequencies by BP stages |
| bp_scatter | Visual | Scatterplot | Scatterplot of BP readings by BP stages |
| dow_tod_plots | Visual | Table Visuals | Counts of BP readings by time of day and day of week |
| bp_ts_plots | Visual | Time Series Plots | BP readings across time |
| bp_report | Report | Summary Report | Combine multiple visuals into a subject-specific report |

## Visualization of blood pressure stages

The blood pressure stages described in the *Classification of blood pressure into stages* section can be visualized using the bp_scatter function. By default, the classification outlined in [9] is used (Table 2); however, the function also allows the user to implement the guidelines from the American Heart Association [23] with automatic adjustment of thresholds depending on bp_type. The threshold values can also be fully customized by the user to match one of the recommended adjustments for out-of-office BP values [26]. Fig 1 highlights the default implementation for subject 70435 of the bp_hypnos data set, with user-specified data segmentation by VISIT and plot wrapping by TIME_OF_DAY. This figure is obtained by the following call:

**Code Block 4**

```
> # Use the hypnos_proc data from the above process_data step

> bp_scatter(hypnos_proc,
             subj = '70435',
             group_var = "VISIT",
             wrap_var = "TIME_OF_DAY")
```

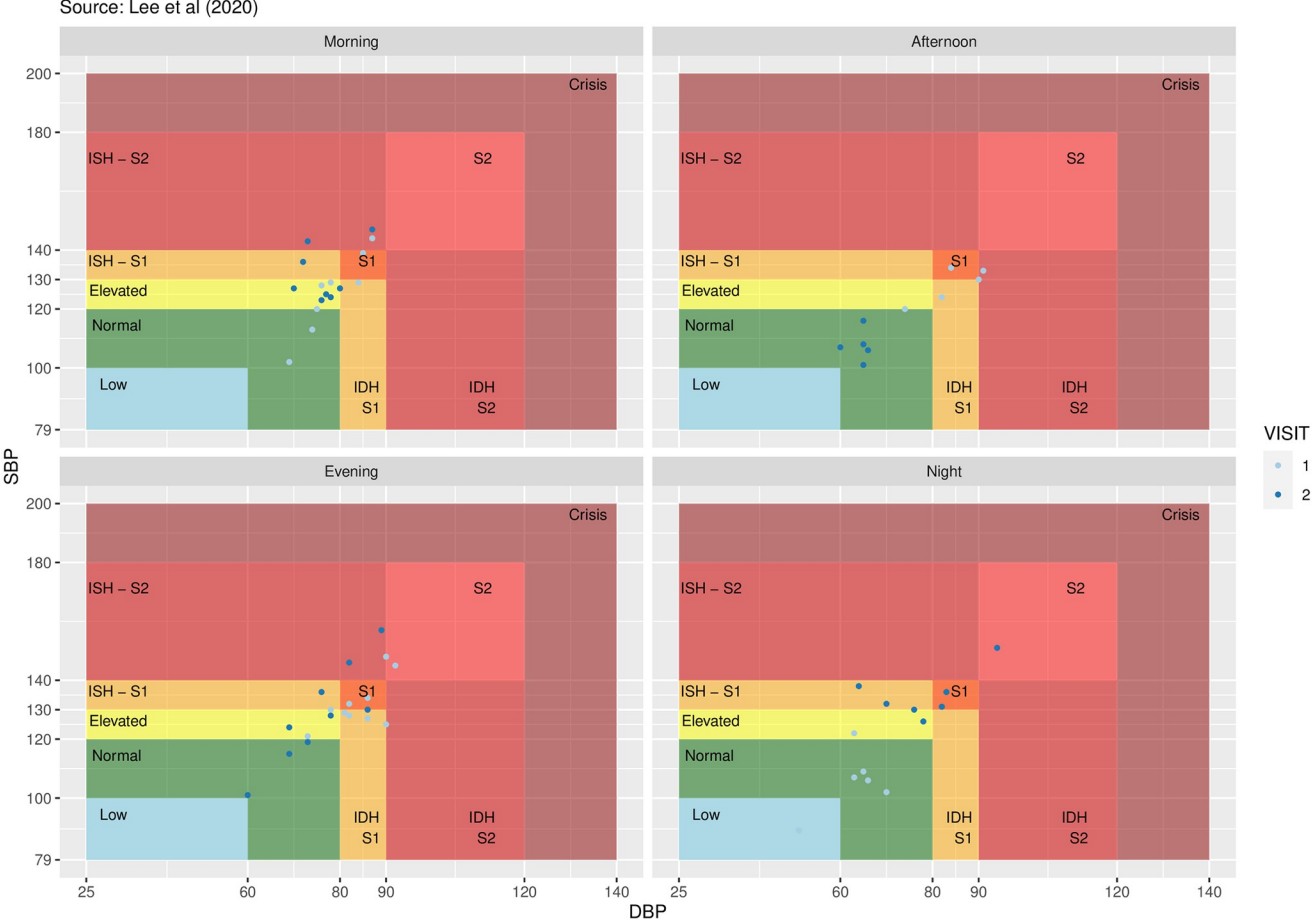

**Fig 1. Scatter plot of blood pressure measurements according to stages for subject 70435 of the `bp_hypnos` data set, the plot is obtained using the `bp_scatter` function segmented by `VISIT` and wrapped by `TIME_OF_DAY`.**

The ability to separate the data by various grouping categories facilitates exploratory analyses; for example, in Fig 1 there is a stark contrast between nighttime BP values during the first visit and nighttime BP values during the second visit.

## Characterizing blood pressure variability

As previously mentioned, blood pressure variability (BPV) plays a crucial role in predicting health outcomes, and characterization of BPV depends on the chosen time-scale: *very short-term* (continuous beat-to-beat arterial BP), *short-term* (within 24 hours), *middle-term* (day-to-day), and *long-term* (visit-to-visit, etc) [3].

The `bp` package contains traditional measures of dispersion, such as standard deviation (SD) and the coefficient of variation (CV), in addition to more modern metrics from the literature, such as the average real variability (`bp_arv`) and successive variation (`bp_sv`). The latter two take into account the temporal information in BP changes, in contrast with SD and CV. ARV is calculated as the average of absolute differences between consecutive readings [13], while SV is based on squared differences [28].

**Code Block 5**

```
> # Calculate ARV for both SBP and DBP using the bp_arv function

> bp_arv(jhs_proc)
# A tibble: 1 x 4
     ID ARV_SBP ARV_DBP     N
* <dbl>   <dbl>   <dbl> <int>
1     1    7.18    6.89   222
```

A comprehensive comparison of all BP variability metrics can be acquired using the `bp_stats` function. Furthermore, the `bp_tables` function offers a list of exploratory tables summarizing frequencies of BP readings by BP stages, time of day, day of the week, etc.

Furthermore, the conventional 24-h standard deviation (SD) fails to account for the contribution of the nocturnal BP fall. The weighted standard deviation (wSD), by contrast, is a combination of standard deviations during the diurnal and nocturnal periods. Thus, it incorporates additional information about the distribution of BP measurements, and has been shown to be associated with ventricular mass index (in contrast to standard SD) [14]. Traditional SD is typically larger than wSD as it does not factor in nocturnal fall. As wSD relies on data during the nocturnal period (i.e. measurements from ABPM), it is calculated within the `bp_sleep_metrics` function.

## Nocturnal BP metrics

Blood pressure follows a circadian variation characterized by fluctuations in a subject's BP levels which rise during the day (diurnal state) when the subject is awake and fall during the night (nocturnal state), typically when the subject is at rest or asleep. This nocturnal BP reduction was first referred to as "dipping" by O'Brien [32] who noted that subjects whose blood pressure does not follow this pattern are at an increased risk for cerebrovascular complications, i.e. stroke.

The `bp` package delineates BP measurements into two states based on the `WAKE` column where `WAKE = 1` corresponds to the diurnal period (awake/daytime), and `WAKE = 0` corresponds to the nocturnal period (asleep/nighttime). It is important to note that in a clinical context, sleep is primarily determined via electroencephalogram (EEG) equipment; for the sake of simplicity however, the `bp` package makes no distinction of whether the sleep indicator in the `WAKE` column is derived from EEG-recorded sleep data or sleep data inferred from actigraphy devices common among most consumer-grade fitness trackers. The `WAKE` column is either specified by the user (e.g. in case this information is available from actigraphy), or is initialized by `process_data` function based on time of day (`ToD`) argument, with night time taken as sleep, and all other times as wake (refer to *Data processing* section).

The percentage of nocturnal decline (known as the dipping percentage) is a BP metric calculated as the percentage change between average BP values during sleep and wake:

$$\text{Dip } \% = \left(1 - \frac{\text{mean } \text{BP}_{SLEEP}}{\text{mean } \text{BP}_{WAKE}}\right) \times 100\%. \tag{1}$$

If Dip% is positive, then BP *decreases*, or "dips", during the night. If Dip% is negative, then BP *increases*, or "rises", nocturnally, which is known as "reverse" dipping.

The Ohasama Study [16] outlines four dipping classification groups: (1) *inverted (or reverse) dippers*, (2) *non-dippers*, (3) *dippers*, and (4) *extreme dippers*. They found that Hypertensive patients tend to have a Dip% < 10%, or a potentially negative percentage indicating reverse dipping. The `bp` package calculates the more recent adaptation of these dipping classification

**Table 4. Dipping classifications and their respective percentage cut-off values according to [15].**

| Dipping Classification | |
|---|---|
| **Dipping Classification** | **Threshold (Dipping%)** |
| *Inverted (Reverse) Dipper* | Dip% ≤ 0% |
| *Non-Dipper* | 0% < Dip% < 10% |
| *Dipper* | 10% ≤ Dip% < 20% |
| *Extreme Dipper* | 20% ≤ Dip% |

groups by Kario et al. [15] along with their corresponding values of Dip% (shown in Table 4) through the `dip_calc` function.

In addition to dipping percentages, the `bp_sleep_metrics` function within the `bp` package allows for computation of other metrics that yield more specific insight into the dynamics of the nocturnal BP behavior. These metrics require the BP readings to be decomposed into the four time slots illustrated in Fig 2 [15]: *presleep* or *evening BP* (all readings 2 hours prior to falling asleep), *lowest BP* (3 readings centered around the lowest BP value during sleep), *prewake BP* (all readings 2 hours prior to waking up), and *postwake* or *morning BP* (all readings 2 hours after waking up). The `bp_sleep_metrics` function will output a list of 4 tables corresponding to count-related summaries of BP data during sleep vs wake (output Table 1), average BP values during each of the time slots illustrated in Fig 2 (output Tables 2 and 3), and a table containing nocturnal BP metrics (output Table 4).

The nocturnal BP metrics include *nocturnal fall* [15], *morningness-eveningness (ME) average / difference* [31], *weighted standard deviation* [14], and the *morning blood pressure surge (MBPS)* metrics [15]. Elevated BP levels during the postwake period are referred to as *morning hypertension* and have been shown to incur the highest incidence of major cardiovascular events [15]. As such, characterizing the blood pressure variability of this period is critical and is often quantified through the morning blood pressure surge in two ways: *sleep-trough MBPS*

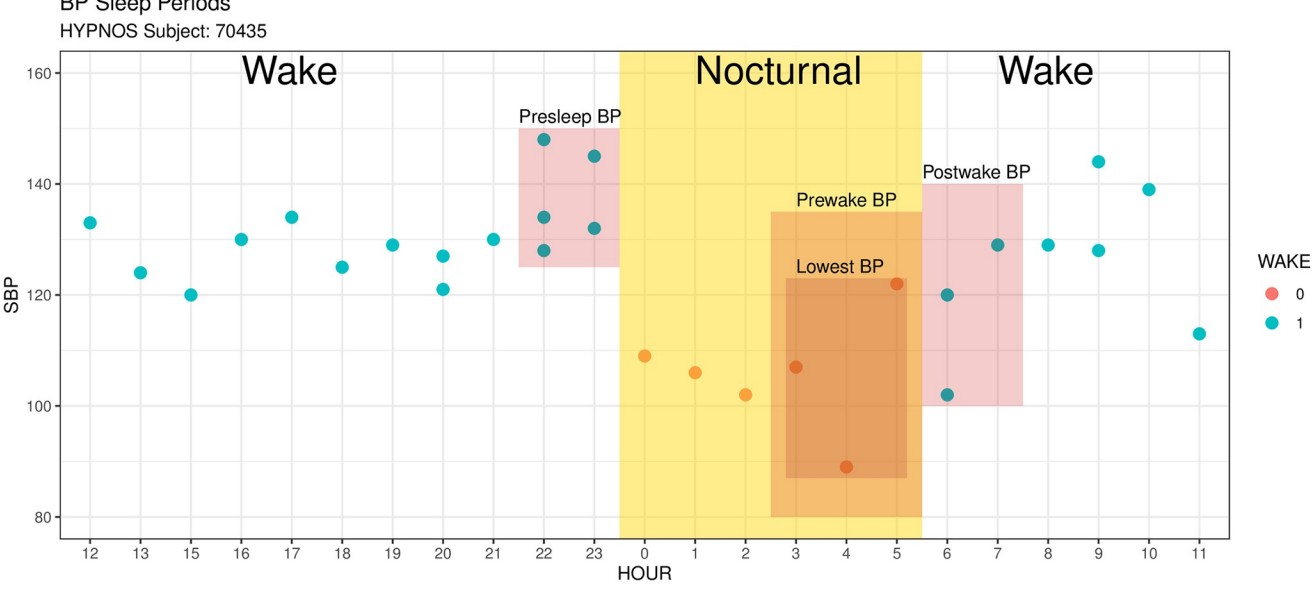

**Fig 2. Decomposition of BP readings into four time slots associated with nocturnal BP readings according to [15].** The delineation into Wake and Sleep is based on time of day (`ToD` argument in `process_data` function), or user-supplied information.

(the difference between the mean BP value during postwake period and the mean BP values over lowest BP period) and *prewake MBPS* (the difference between the mean BP values during the postwake period and the mean BP values during the prewake period). The corresponding output from bp_sleep_metrics includes the following columns: dip_calc_* (nocturnal dipping), noct_fall_* (nocturnal fall), ST_mbps_* (sleep-trough MBPS), PW_mbps_* (prewake MBPS), ME_*_avg (ME average), ME_*_diff (ME difference) and wSD_* (weighted standard deviation), where * is either SBP or DBP. Collectively, these metrics fully characterize the behavior of an individual's BP before, during, and after the nocturnal period.

## Case Study I—Single subject HBPM data

HBPM data have a higher degree of measurement error as a result of the user initializing the BP monitoring device [33] compared to ABPM data which automatically takes recordings at prescribed intervals. Here, we consider the bp_jhs data set [20] which corresponds to a single-subject HBPM data set, wherein the user took multiple measurements in succession within short intervals of time. The user also exhibited atypical circadian rhythms (falling asleep well after midnight). We show how the bp package can be used to automatically process the data, aggregate the consecutive measurements, adjust the day-night allocation based on individual circadian rhythms, and create a one page report for easy communication of HBPM summaries.

### Aggregation of consecutive readings

Schulze at al. [34] argue that the number of HBPM BP readings taken in succession along with the nature of how the readings were taken (left vs. right arm, supine vs. sitting vs. standing, etc.) significantly influences the variation of the resulting BP measurements. Furthermore, the longer one sits to take additional readings, the more one's BP decreases due to the natural decline in BP over prolonged sitting time. Therefore, it is recommended to average successive readings to reduce measurement error.

In the case of bp_jhs, there are many instances where readings were taken in succession (generally within 3 minutes), but the subject was not consistent in taking the same number of successive readings at the same times throughout each day. A subset of the data *without* averaging is shown below.

**Code Block 6**

```
> # Run the process_data function using the original bp_jhs data set to
> # create a new processed data set called jhs_proc

> jhs_proc <- process_data(bp_jhs, bp_type = "hbpm",
                                    sbp = "Sys.mmHg.",
                                    dbp = "Dias.mmHg.",
                           date_time = "DateTime",
                                    hr = "Pulse.bpm.")

> # Take a small sample to explore

> jhs_proc[4:14, c("SBP", "DBP", "BP_CLASS", "ID", "GROUP", "DATE_TIME",
                    "DATE", "DAY_OF_WEEK", "HOUR", "TIME_OF_DAY")]

    SBP DBP BP_CLASS ID GROUP        DATE_TIME       DATE DAY_OF_WEEK HOUR TIME_OF_DAY
4   130  81  Stage 1  1     1 2019-07-30 13:47:46 2019-07-30         Tue   13   Afternoon
5   134  83  Stage 1  1     1 2019-07-30 13:46:15 2019-07-30         Tue   13   Afternoon
6   140  84 ISH - S2  1     1 2019-07-29 11:18:13 2019-07-29         Mon   11     Morning
7   132  79 ISH - S1  1     1 2019-07-29 05:44:57 2019-07-29         Mon    5       Night
8   126  80 IDH - S1  1     1 2019-07-29 05:43:23 2019-07-29         Mon    5       Night
9   142  82 ISH - S2  1     1 2019-07-29 05:41:35 2019-07-29         Mon    5       Night
10  144  86 ISH - S2  1     1 2019-07-28 15:18:07 2019-07-28         Sun   15   Afternoon
11  150  92  Stage 2  1     1 2019-07-28 15:16:29 2019-07-28         Sun   15   Afternoon
12  140  88 ISH - S2  1     1 2019-07-28 02:05:02 2019-07-28         Sun    2       Night
13  144  90  Stage 2  1     1 2019-07-28 01:58:16 2019-07-28         Sun    1       Night
14  137  85  Stage 1  1     1 2019-07-26 12:59:10 2019-07-26         Fri   12   Afternoon
```

There are three separate instances of consecutive readings in the above output: two recordings on Sunday around 3:15PM (rows 10—11) within two minutes of each other, three recordings on Monday around 5:40AM (rows 7—9) within three minutes of each other, and two recordings on Tuesday around 1:45PM (rows 4—5) within two minutes of each other.

The `process_data` function automatically aggregates BP measurements using the `agg = TRUE` option with a user-specified threshold (`agg_thresh`) denoting the maximum duration (in minutes) between successive readings, with the default value being 3 minutes.

**Code Block 7**

```
> # Set 'agg = TRUE' in order to aggregate consecutive observations

> jhs_proc_agg <- process_data(bp_jhs, bp_type = "hbpm",
                                        sbp = "Sys.mmHg.",
                                        dbp = "Dias.mmHg.",
                               date_time = "DateTime",
                                        hr = "Pulse.bpm.",
                                       agg = TRUE)

> jhs_proc_agg[4:14, c("SBP", "DBP", "BP_CLASS", "ID", "GROUP",
                "DATE_TIME", "DATE", "DAY_OF_WEEK", "HOUR", "TIME_OF_DAY")]

    SBP DBP BP_CLASS ID GROUP        DATE_TIME       DATE DAY_OF_WEEK HOUR TIME_OF_DAY
4   132  82  Stage 1  1     1 2019-07-30 13:47:46 2019-07-30         Tue   13   Afternoon
5   132  82  Stage 1  1     1 2019-07-30 13:46:15 2019-07-30         Tue   13   Afternoon
6   140  84 ISH - S2  1     1 2019-07-29 11:18:13 2019-07-29         Mon   11     Morning
7   133  80  Stage 1  1     1 2019-07-29 05:44:57 2019-07-29         Mon    5       Night
8   133  80  Stage 1  1     1 2019-07-29 05:43:23 2019-07-29         Mon    5       Night
9   133  80  Stage 1  1     1 2019-07-29 05:41:35 2019-07-29         Mon    5       Night
10  147  89 ISH - S2  1     1 2019-07-28 15:18:07 2019-07-28         Sun   15   Afternoon
11  147  89 ISH - S2  1     1 2019-07-28 15:16:29 2019-07-28         Sun   15   Afternoon
12  140  88 ISH - S2  1     1 2019-07-28 02:05:02 2019-07-28         Sun    2       Night
13  144  90  Stage 2  1     1 2019-07-28 01:58:16 2019-07-28         Sun    1       Night
14  141  85 ISH - S2  1     1 2019-07-26 12:59:10 2019-07-26         Fri   12   Afternoon
```

In the output above, all instances of consecutive readings within 3 minutes of each other have been averaged together as in the case of rows 4—5, rows 7—9, and rows 10—11. Previously, the BP stages in the original data set classified row 7 as *ISH—S1*, row 8 as *IDH—S1* and row 9 as *ISH—S2*. After aggregation, the readings for this particular instance are all reclassified to *Stage 1* based on their new average values (133 SBP / 80 DBP). By default, the number of rows stays in tact representing the original data, however the user can further collapse the data set to omit redundant rows using the `collapse_df` argument.

**Code Block 8**

```
> # To avoid having repeated values from the aggregation, set
> # collapse_df = 'TRUE' in order to reduce the data set and omit
> # repetitions

> jhs_proc_agg_collapsed <- process_data(bp_jhs, bp_type = "hbpm",
                                          sbp = "Sys.mmHg.",
                                          dbp = "Dias.mmHg.",
                                     date_time = "DateTime",
                                            hr = "Pulse.bpm.",
                                           agg = TRUE,
                                   collapse_df = TRUE)

> jhs_proc_agg_collapsed[3:9, c("SBP", "DBP", "BP_CLASS", "ID", "GROUP",
           "DATE_TIME", "DATE", "DAY_OF_WEEK", "HOUR", "TIME_OF_DAY")]
  SBP DBP BP_CLASS ID GROUP      DATE_TIME       DATE DAY_OF_WEEK HOUR TIME_OF_DAY
3 132  82  Stage 1  1     1 2019-07-30 13:47:46 2019-07-30     Tue   13   Afternoon
4 140  84 ISH - S2  1     1 2019-07-29 11:18:13 2019-07-29     Mon   11     Morning
5 133  80  Stage 1  1     1 2019-07-29 05:44:57 2019-07-29     Mon    5       Night
6 147  89 ISH - S2  1     1 2019-07-28 15:18:07 2019-07-28     Sun   15   Afternoon
7 140  88 ISH - S2  1     1 2019-07-28 02:05:02 2019-07-28     Sun    2       Night
8 144  90  Stage 2  1     1 2019-07-28 01:58:16 2019-07-28     Sun    1       Night
9 141  85 ISH - S2  1     1 2019-07-26 12:59:10 2019-07-26     Fri   12   Afternoon
```

As shown above, when `collapse_df = TRUE`, the first row of each consecutive reading sequence is kept and the redundant rows (in this case, rows 5, 8, 9, and 11 from the previous output) that remain are removed, reducing the overall size of the resulting data set.

## End-of-Day (EOD) determination

Depending on the population of interest, the delineation between night and day may change due to late-night work shifts, time zone differences, or individuals' circadian rhythms. In the case of the `bp_jhs` data set, the subject almost never fell asleep before midnight, with the sleep onset times being highly variable (from before midnight to 5AM). Given that the data is of the HBMP type, it is clear that each reading occurred when the subject was awake; however, summarizing such data using the standard midnight cutoff for end-of-day may not be advisable. Depending on the context, readings could be considered as part of either the previous day's readings (in the event that the reading was taken very late and the subject had not yet gone to sleep) or the current day's readings (in the case that the subject woke up very early).

Notice that in the original (non-collapsed) subset of the `bp_jhs` data from the *Aggregation of consecutive readings* section, the consecutive reading sequence on Monday night (rows 7—9) as well as the non-consecutive readings on Sunday night (rows 12 and 13) are all recorded in the early hours of the day (around 5AM and 2AM, correspondingly). These readings are thus either a part of the previous day's readings (i.e. the reading was taken very late and the subject had not yet gone to sleep), or the current day's readings (i.e. the subject woke up very early). By default, all readings will be assigned to the current date; however, this can be adjusted by the user by supplying the optional end-of-day argument (`eod`) in the

`process_data` function. Here, we set the cutoff at 6AM, so that all readings in the early morning hours are automatically allocated to the previous day.

**Code Block 9**

```
> # Here, we can specify a specific end-of-day time by which we delineate
> # one day from another
> jhs_proc_eod <- process_data(bp_jhs, bp_type = ’hbpm’,
                                          sbp = "Sys.mmHg.",
                                          dbp = "Dias.mmHg.",
                                    date_time = "DateTime",
                                           hr = "pulse.bpm.",
                                          eod = "0600")

> jhs_proc_eod[4:14, c("SBP", "DBP", "BP_CLASS", "ID", "GROUP", "DATE_TIME",
                       "DATE", "DAY_OF_WEEK", "HOUR", "TIME_OF_DAY")]

    SBP DBP BP\_CLASS ID GROUP           DATE_TIME       DATE DAY_OF_WEEK HOUR TIME_OF_DAY
4   130  81  Stage 1  1     1 2019-07-30 13:47:46 2019-07-30         Tue   13   Afternoon
5   134  83  Stage 1  1     1 2019-07-30 13:46:15 2019-07-30         Tue   13   Afternoon
6   140  84 ISH - S2  1     1 2019-07-29 11:18:13 2019-07-29         Mon   11     Morning
7   132  79 ISH - S1  1     1 2019-07-29 05:44:57 2019-07-28         Sun    5       Night
8   126  80 IDH - S1  1     1 2019-07-29 05:43:23 2019-07-28         Sun    5       Night
9   142  82 ISH - S2  1     1 2019-07-29 05:41:35 2019-07-28         Sun    5       Night
10  144  86 ISH - S2  1     1 2019-07-28 15:18:07 2019-07-28         Sun   15   Afternoon
11  150  92  Stage 2  1     1 2019-07-28 15:16:29 2019-07-28         Sun   15   Afternoon
12  140  88 ISH - S2  1     1 2019-07-28 02:05:02 2019-07-27         Sat    2       Night
13  144  90  Stage 2  1     1 2019-07-28 01:58:16 2019-07-27         Sat    1       Night
14  137  85  Stage 1  1     1 2019-07-26 12:59:10 2019-07-26         Fri   12   Afternoon
```

Here the `DATE` and `DAY_OF_WEEK` columns are adjusted, e.g. for rows 12-13 the allocated date now shows 2019-07-27 instead of 2019-07-28 and the day of week is Saturday instead of Sunday.

## Generating a report

The `bp` package is equipped with a report generating function, `bp_report`, that offers a high-level overview of trends pertaining to an individual's BP patterns.

The following code generates a report for the `bp_jhs` data set after aggregating the data using the 3 minute threshold (refer to *Aggregation of consecutive readings* section) and setting the end-of-day delineation to 6AM (refer to *End-of-Day (EOD) Determination* section).

**Code Block 10**

```
> # Create a processed dataframe that incorporates all
> # of the necessary adjustments:
> # - end-of-day (eod) delineation
> # - aggregation of consecutive measurements (agg)
> # - omission of redundant rows (collapse_df)

> jhs_proc_report <- process_data(bp_jhs, bp_type = ’hbpm’,
                                          sbp = "Sys.mmHg.",
                                          dbp = "Dias.mmHg.",
                                    date_time = "DateTime",
                                           hr = "pulse.bpm.",
                                          eod = "0600",
                                          agg = TRUE,
                                  collapse_df = TRUE)

> bp_report(jhs_proc_report, group_var = "TIME_OF_DAY")
```

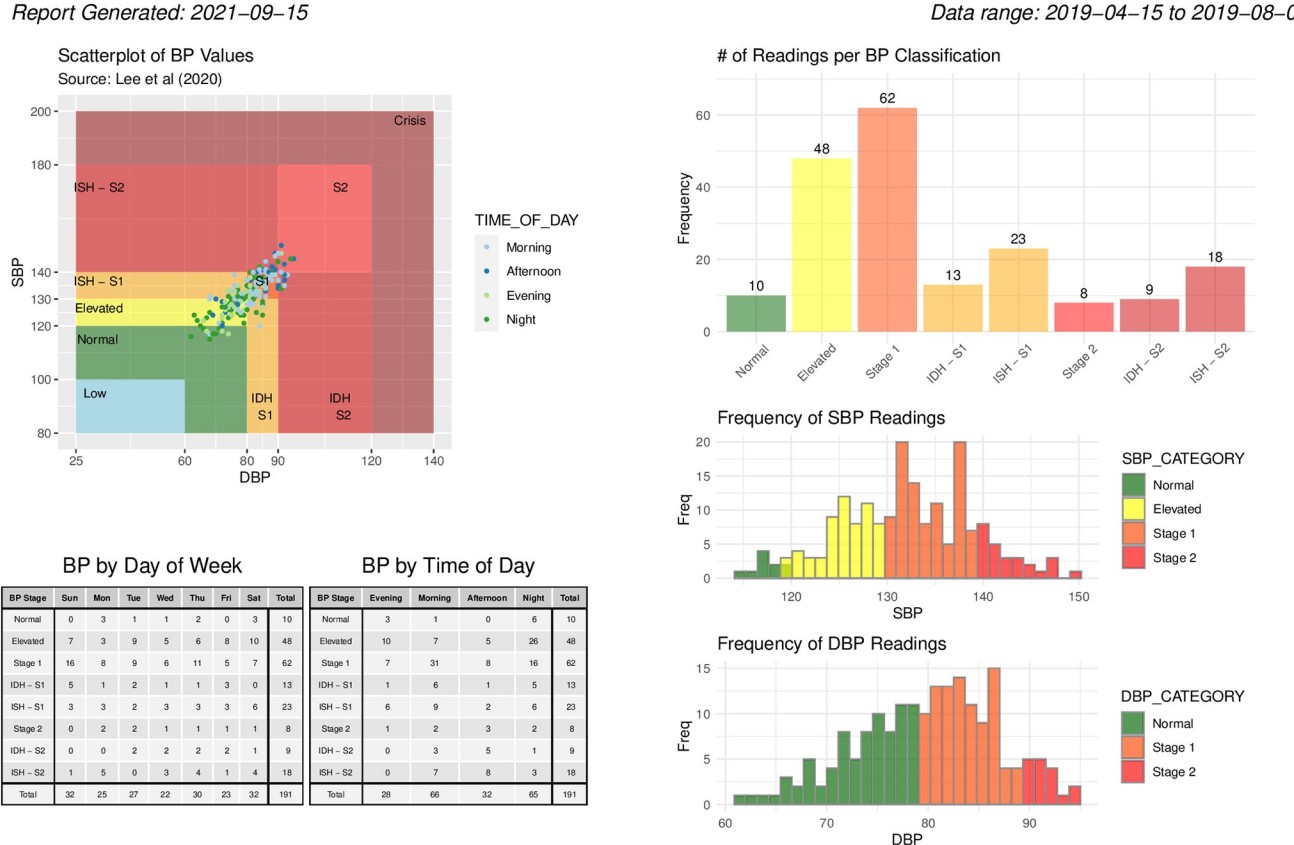

**Fig 3. A BP report for the `bp_jhs` data set generated using the `bp_report` function.**

Fig 3 shows the generated report that illustrates the frequency of BP readings by stage classification, and additional stratification of scatter plot by a given categorical variable (`TIME_OF_DAY`). From the scatter plot, BP values are concentrated around the center and do not exhibit large fluctuations. From the table that breaks BP down by `TIME_OF_DAY`, it is clear that most of the readings occurred at night, with night also having lower readings compared to other times of day.

## Case Study II—Multi-subject ABPM data

In contrast to HBPM data, ABPM data offers insight into an individual's nocturnal BP pattern. Here, we consider `bp_hypnos` data set which has 24-hour ABPM measurements for multiple subjects over two separate visits. We show how the `bp` package can be used to visualize the data across visits and hours of the day to illustrate the nocturnal BP fall as well as identify potential outliers, show how ABPM data can be used for dipping classification (as in Table 4), and show how BP variability metrics that account for nocturnal fall, i.e. weighted SD [14], differ from conventional overall standard deviation.

### Time series plots

The `bp_hypnos` dataset contains blood pressure measurements of multiple subjects, where each subject obtained ABPM readings during two separate visits. The `bp_ts_plots`

function allows the user to visually compare blood pressure changes over time for selected subjects, and stratify the comparisons through an optional grouping factor (e.g. VISIT in the case of bp_hypnos).

**Code Block 11**

```
> # Create a separate variable to store time series plots, specifying:
> #  - the first hour is 11 (11:00),
> #  - the plots be wrapped by visit number (either 1 or 2 in this case),
> #  - the data correspond to subjects 70435 and 70439
> out <- bp_ts_plots(hypnos_proc, first_hour = 11,
                                  wrap_var = 'visit',
                                  subj = c('70435', '70439') )
```

The resulting out object is a list of length two containing (i) a list of subject-specific time series plots separated by visit (Fig 4, top); (ii) a list of subject-specific plots of BP measurements by hour across all visits (Fig 4, bottom).

The plots in Fig 4 reveals that BP patterns are quite different across subjects; subject 70435 has fairly large fluctuations in both SBP and DBP values throughout the 24 hour period for each of the visits, whereas subject 70439 has almost no variation during the second visit. To view the nocturnal hours more easily, the first_hour argument is utilized to shift the x-axis in the hour plot to begin at 11 AM. While subject 70435 demonstrates expected BP

(a)

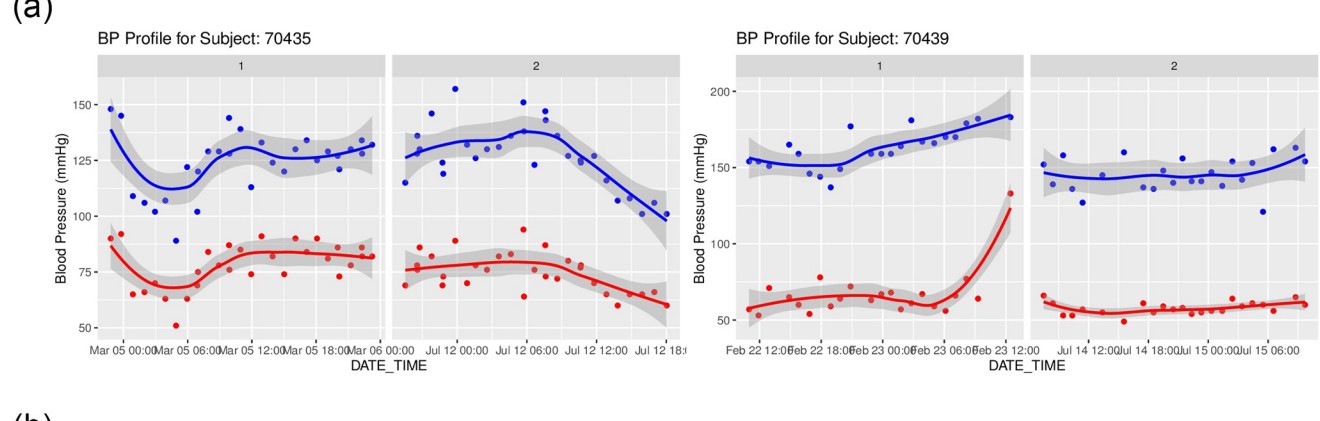

(b)

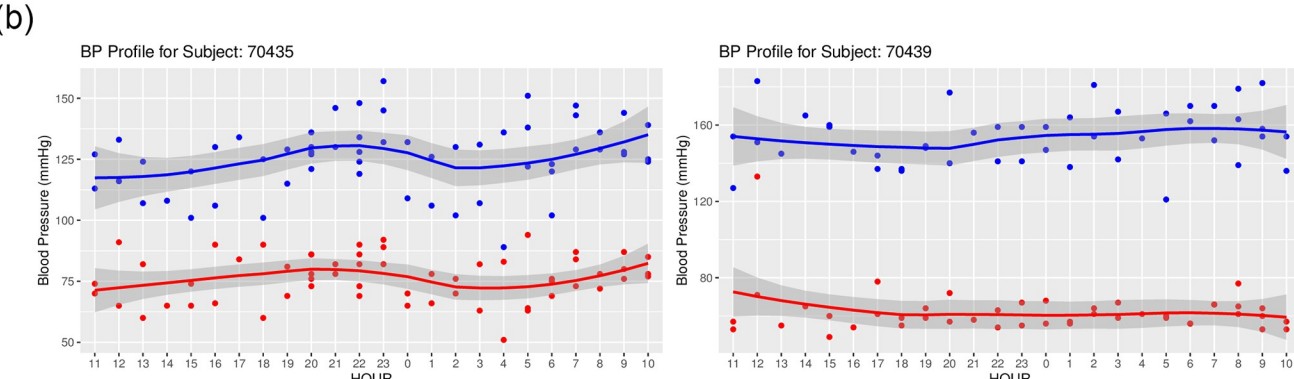

**Fig 4. Time series plots for subjects 70435 and 70439 from the bp_hypnos data set separated by visit (top row), and by hour across all visits (bottom row).** All plots are generated using the bp_ts_plots function in the bp package.

fluctuations (decrease during nighttime hours and rise during daytime activity), the BP of subject 70439 remains relatively flat throughout the day.

## Nocturnal dipping calculation

To further quantify the observed differences in nocturnal BP fluctuations for both subjects, we calculate dipping percentages and perform dipping classification as described in the *Nocturnal BP metrics* section using function `dip_calc`.

**Code Block 12**

```
> # Dipping calculation for subject 70435:

> dip_calc(hypnos_proc, subj = '70435')

[[1]]
# A tibble: 4 x 6
# Groups:   ID, VISIT [2]
  ID    VISIT  WAKE avg_SBP avg_DBP     N
  <fct> <dbl> <dbl>   <dbl>   <dbl> <int>
1 70435     1     0    106.      63     6
2 70435     1     1    129.    82.1    23
3 70435     2     0    136.    79.2     9
4 70435     2     1    123.    72.5    20

[[2]]
# A tibble: 2 x 6
# Groups:   ID [1]
  ID    VISIT dip_sys class_sys dip_dias class_dias
  <fct> <dbl>   <dbl> <chr>        <dbl> <chr>
1 70435     1   0.179 dipper       0.233 extreme
2 70435     2  -0.104 reverse    -0.0927 reverse

> # Dipping calculation for subject 70439:

> dip_calc(hypnos_proc, subj = '70439')

[[1]]
# A tibble: 4 x 6
# Groups:   ID, VISIT [2]
  ID    VISIT  WAKE avg_SBP avg_DBP     N
  <fct> <dbl> <dbl>   <dbl>   <dbl> <int>
1 70439     1     0     167    62.6     8
2 70439     1     1    160.    69.3    14
3 70439     2     0    149.    60.8     6
4 70439     2     1    144.    56.8    17

[[2]]
# A tibble: 2 x 6
# Groups:   ID [1]
  ID    VISIT dip_sys class_sys dip_dias class_dias
  <fct> <dbl>   <dbl> <chr>        <dbl> <chr>
1 70439     1 -0.0442 reverse     0.0961 non-dipper
2 70439     2 -0.0329 reverse    -0.0717 reverse
```

The function returns two tables. The first table returns average SBP and DBP values broken down by WAKE status (where 0 indicates nocturnal measurements, i.e. the sleep period, and 1 indicates diurnal measurements, i.e. the wake period), and groups such as VISIT (if applicable). The second table returns dipping percentages (in proportion format) and corresponding dipping classifications according to Table 4.

From the output, subject 70435 exhibits pronounced decreases in nocturnal BP during the first visit ("dipper" and "extreme" classifications for SBP and DBP, respectively); however, both SBP and DBP show a reverse dipping pattern for the subject's second visit. In contrast, subject 70439 has similar dipping percentages for SBP across both visits (with the consistent "reverse" dipping classification), with DBP classification shifting from the "non-dipper" to "reverse" dipper category. Note also that the overall BP averages for subject 70439 decreased from first to second visit (the avg_SBP and avg_DBP columns) creating a contrast between the overall lowering of BP between visits and the worsening of the nocturnal patterns of the individual's DBP dipping behavior.

Examining the time series plot for subject 70439 (Fig 4) in conjunction with the output of dip_calc reveals an apparent DBP measurement at the end of the first visit that is an outlier. This is supported by investigating the top three highest DBP values for this subject using the dplyr package [35]:

**Code Block 13**

```
> # Explore the top 3 highest DBP values
> # Arrange DBP in descending order

> hypnos_proc %>%
        filter(ID =='70439') %>%
        arrange(desc(DBP)) %>%
        head(3) %>%
        dplyr::select(SBP, DBP, BP_CLASS, DATE_TIME, MAP, HR,
                      RPP, WAKE, VISIT)

  SBP DBP BP_CLASS          DATE_TIME      MAP  HR   RPP WAKE VISIT
1 183 133   Crisis 2017-02-23 12:27:00 149.6667 101 18483    1     1
2 144  78 ISH - S2 2017-02-22 17:54:00 100.0000  63  9072    1     1
3 179  77 ISH - S2 2017-02-23 08:09:00 111.0000  60 10740    1     1
```

The largest DBP value is nearly double the second highest value; a clear indication of an outlier. bp allows automatic filtering for such outliers in the process_data function via DUL argument (DBP Upper Limit). By default, the argument is set to 140 in accordance with [36]; however, to remove the outlier for subject 70439, we set DUL = 130 and recalculate the dipping percentages.

**Code Block 14**

```
> # Re-process data according to a DUL threshold of 130

> hypnos_proc_recalc <- process_data(bp_hypnos, bp_type = "abpm",
                                                 sbp = "SYST",
                                                 dbp = "DIAST",
                                           date_time = "DATE.TIME",
                                                  hr = "HR",
                                                  pp = "PP",
                                                 map = "MAP",
                                                 rpp = "RPP",
                                                  id = "ID",
                                               visit = "VISIT",
                                                wake = "WAKE",
                                                 DUL = 130)

> dip_calc(hypnos_proc_recalc, subj = '70439' )

[[1]]
# A tibble: 4 x 6
# Groups:   ID, VISIT [2]
   ID    VISIT WAKE  avg_SBP avg_DBP     N
   <fct> <fct> <fct>   <dbl>   <dbl> <int>
1 70439 1     0         167    62.6     8
2 70439 1     1         158.   64.4    13
3 70439 2     0         149.   60.8     6
4 70439 2     1         144.   56.8    17

[[2]]
# A tibble: 2 x 6
# Groups:   ID [1]
   ID    VISIT dip_sys class_sys dip_dias class_dias
   <fct> <fct>   <dbl> <chr>        <dbl> <chr>
1 70439 1     -0.0559 reverse     0.0273 non-dipper
2 70439 2     -0.0329 reverse    -0.0717 reverse
```

Eliminating the outlier in `hypnos_proc` during the first visit decreased the overall average SBP and DBP values in the `hypnos_proc_recalc` output during the wake period (specifically, from 160 to 158 for `avg_SBP` and from 69.3 to 64.4 for `avg_DBP` in output `[[1]]` when `VISIT = 1` and `WAKE = 1`). This led to a decrease in the percentage that SBP fell during the nocturnal period from -4.42% to -5.59% (more precisely, an increase in the percentage that SBP rose during the nocturnal period) and a decrease in the percentage that DBP fell during the nocturnal period from +9.61% to +2.73% as a result of a smaller magnitude difference between values during wake and nocturnal periods. Note that despite the adjustments to these dipping percentages, the dipping classifications remained the same.

The function `dip_class_plot` provides a display of dipping percentages together with their classification in accordance with [16], and can be used to visualize the effect of removing the outlier by considering both the original data and the data after processing.

**Code Block 15**

```
# Outlier included
dip_class_plot(hypnos_proc, subj = c('70435', '70439'))

# Outlier omitted
dip_class_plot(hypnos_proc_recalc, subj = c('70435', '70439'))
```

Fig 5 shows both outputs from Code block 15, with the left plot corresponding to `hyp-nos_proc` and the right plot corresponding to `hypnos_proc_recalc`.

## Standard deviation (SD) vs weighted standard deviation (wSD)

As noted earlier in the *Characterizing blood pressure variability* section, conventional 24-h standard deviation (SD) fails to account for the contribution of the nocturnal BP fall and is typically larger than wSD. This is reflected for subject 70439 in `hypnos_proc` data (original data with outlier).

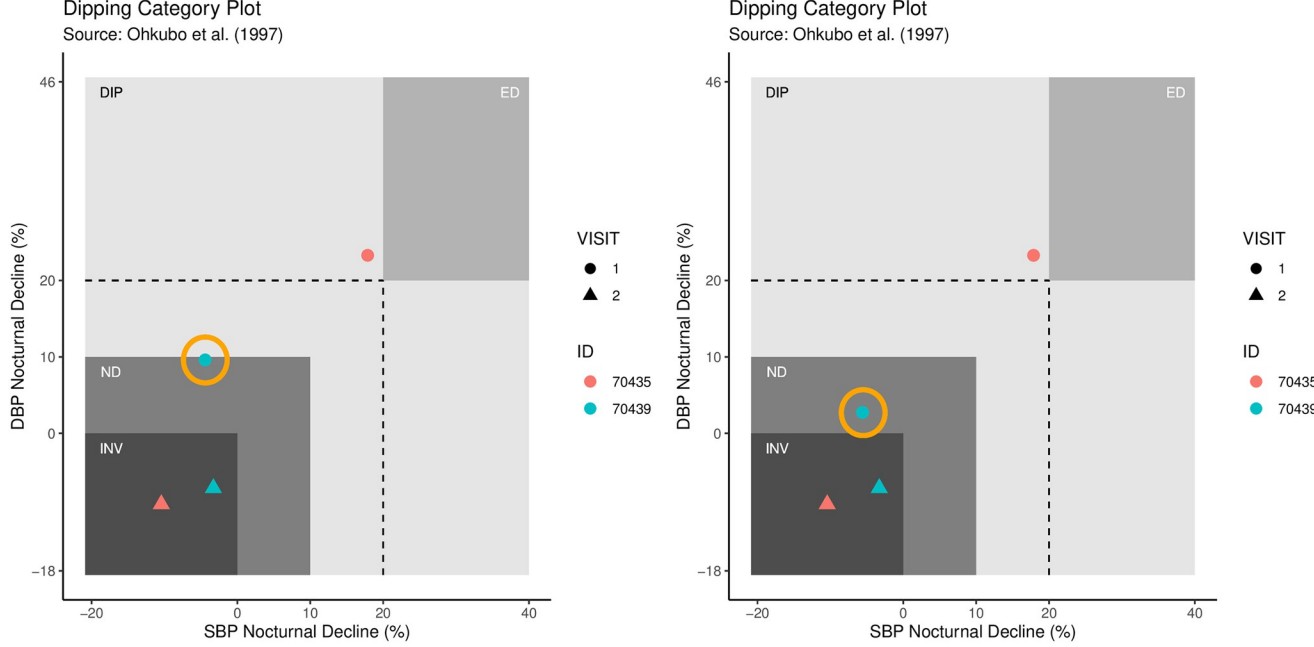

**Fig 5. Output of `dip_class_plot` function on selected subjects from `bp_hypnos` data.** Left plot shows dip percentages using the original data. Right plot shows dip percentages using the processed data with one outlier removed for subject 70439.

**Code Block 16**

```
> # Run the bp_sleep_metrics and bp_stats functions using
> # the original hypnos_proc data to obtain SD and wSD

> cbind(
+   bp_sleep_metrics(hypnos_proc, subj = '70439')[[4]][,c(1:2, 10, 17)],
+   bp_stats(hypnos_proc, subj = '70439', inc_wake = F)[,c(8:9)]
+   )

# A tibble: 2 x 6
# Groups:   ID, VISIT [2]
  ID    VISIT wSD_SBP wSD_DBP SD_SBP SD_DBP
  <fct> <fct>   <dbl>   <dbl>  <dbl>  <dbl>
1 70439 1        12.0    14.2   13.0   16.3
2 70439 2        11.0     3.75  11.0    4.12
```

We now consider how SD and wSD change when the outlier is removed (via the `hyp-nos_proc_recalc` data).

**Code Block 17**

```
> # Re-run using hypnos_proc_recalc data instead

> cbind(
+   bp_sleep_metrics(hypnos_proc_recalc, subj = '70439')[[4]][,c(1:2, 10, 17)],
+   bp_stats(hypnos_proc_recalc, subj = '70439', inc_wake = F)[,c(8:9)]
+ )

# A tibble: 2 x 6
# Groups:   ID, VISIT [2]
  ID    VISIT wSD_SBP wSD_DBP SD_SBP SD_DBP
  <fct> <fct>   <dbl>   <dbl>  <dbl>  <dbl>
1 70439 1        12.0     6.90   12.5   6.98
2 70439 2        11.0     3.75   11.0   4.12
```

As expected, the removal of the outlying measurement as described in the *Nocturnal dipping calculation* section reduces both metrics of standard deviation (SD and wSD); however, the conventional SD still remains greater than wSD.

## Discussion and future directions

The `bp` package is designed to simplify the analysis of blood pressure data, and assist users with data processing, data visualization, and BP metric calculations following recommendations set forth by O'Brien and Atkins [27]. The package includes a variety of sample data sets as references that cover all BP data types (AP, HBPM, and ABPM) to help get started. New BP metrics and visualizations will be incorporated into future versions of `bp` as they develop. The free and open-source nature of R makes the `bp` package advantageous compared with existing proprietary commercial software (e.g. AccuWin Pro™ by SunTech Medical, CardioSoft™ by GE Healthcare) and further allows for the creation of reproducible scripts that document the full analysis workflow from data processing to visualizations. A current limitation is that the users are required to have R programming experience, which may be a drawback for researchers and clinicians who are accustomed to point and click graphical user interfaces. Ideally, we would like to have a software solution that combines the advantages of R language with

graphical user interface. This is possible to achieve by developing an accompanying Shiny application [37], an interactive web application that provides graphical user interface for underlying R scripts. It is our goal to develop such a Shiny App for the **bp** package to further increase package accessibility in future work. Another limitation is the lack of device-specific import capabilities. An ideal configuration will allow to load the data directly from the device into R, however this will likely necessitate substantial adjustments to application programming interface (API) of the device or associated app, falling outside the scope of R package capabilities. An alternative is to work directly with existing export capabilities of the devices, most of which allow to export the data into CSV format. Such format is easy to load into R, and any device-specific discrepancies in column names, time configurations or pre-calculated valuates are standardized by `process_data` function as long as two basic inputs (SBP and DBP) are specified. Nevertheless, continuous development of new ABPM devices coupled with potentially varying data export formats across devices present definite challenges for any blood pressure analyses software. Finally, it would be of great interest to compare custom `bp` functionalities like dipping calculation in `dip_calc` against other software solutions, however such cross-comparison is challenging due to the lack of custom BP functionalities in SAS and SPSS software, and the cost associated with obtaining proprietary BP software (e.g. Accu-Win Pro™ by SunTech Medical). Nevertheless, we hope that public availability of our example datasets together with underlying source code is a useful preliminary step toward this endeavor.

## Acknowledgments

The authors are thankful to Elizabeth Chun for her help in streamlining the calculations in `bp_sleep_metrics` function, and to Devon Maywald for his feedback on user experience with the earlier versions of the package.

## Author Contributions

**Conceptualization:** John Schwenck, Irina Gaynanova.

**Data curation:** John Schwenck, Naresh M. Punjabi.

**Formal analysis:** John Schwenck.

**Methodology:** John Schwenck.

**Project administration:** Irina Gaynanova.

**Software:** John Schwenck, Irina Gaynanova.

**Supervision:** Irina Gaynanova.

**Validation:** John Schwenck, Irina Gaynanova.

**Visualization:** John Schwenck.

**Writing – original draft:** John Schwenck.

**Writing – review & editing:** John Schwenck, Naresh M. Punjabi, Irina Gaynanova.

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
