## [Decision Letter · Decision Letter 0]

29 Jun 2022

PONE-D-22-13672bp: Blood Pressure Analysis in RPLOS ONE

Dear Dr. Irina Gaynanova,

Thank you for submitting your manuscript to PLOS ONE. After careful consideration, we feel that it has merit but does not fully meet PLOS ONE’s publication criteria as it currently stands. Therefore, we invite you to submit a revised version of the manuscript that addresses the points raised during the review process. In my opinion the work addresses an important topic and it should not be difficult to address all  points raised by the reviewers.

We look forward to receiving your revised manuscript.

Kind regards,

Fabiana Zama

Academic Editor

PLOS ONE

Journal Requirements:

Reviewers' comments:

Reviewer's Responses to Questions

**Comments to the Author**

1. Is the manuscript technically sound, and do the data support the conclusions?

Reviewer #1: Partly

Reviewer #2: Yes

2. Has the statistical analysis been performed appropriately and rigorously? 

Reviewer #1: No

Reviewer #2: Yes

3. Have the authors made all data underlying the findings in their manuscript fully available?

Reviewer #1: Yes

Reviewer #2: Yes

4. Is the manuscript presented in an intelligible fashion and written in standard English?

Reviewer #1: Yes

Reviewer #2: Yes

5. Review Comments to the Author

Reviewer #1: In this paper, the authors present a program for analyzing continuous BP, home BP, and ambulatory BP data. The program is developed in R language and provides data analysis and sample datasets functions.

In general, although not novel, the idea is interesting since the software is provided as an open-source program.

However, as correctly stated by the authors, the user needs to know how the R language works.

Usually, BP data sets are analyzed with statistical packages such as SAS and SPSS. These programs are not free, at variance with the proposed program. However, they are beneficial in analyzing BP data by constructing simple lines of code. The authors should discuss the advantage of their program compared to these packages. The authors should have validated results obtained with their code with any known validated and certified software.

A significant problem with this program is the classification of BP values which is based on office BP that are applied to home and ambulatory BP that have different thresholds. Since they are statisticians, I recommend they seek the support of an expert in out-of-office BP analysis. They should also refer to significant guidelines such as those from ESH: home BP https://doi.org/10.1097/HJH.0000000000002922

ambulatory BP https://doi.org/10.1097/HJH.0000000000000221

I recommend moving the explanation regarding wSD to the section “Characterizing blood pressure variability,” to which it better belongs.

Reviewer #2: The work presented by Schwenck et al. addresses the unmet need for an open-source environment that is able to perform a large variety of blood pressure-related calculations in R independent from the device manufacturer. The manuscript is very well written, contains an adequate amount of references for all employed protocols and calculations and gives plenty of examples on how to use the software. I only have a few minor comments for the authors.

1) I might have missed it, but the open source license of the software should be mentioned in the manuscript.

2) Code listings from e.g. from line 508 on could include code comment blocks such as “# here we load the data” or similar as this makes the blocks easier to read and may open the software to novice R users.

3) For the final version of the manuscript it may also be beneficial to add some kind of label to code blocks in order to refer to them from within the text.

4) Although this goes more in the direction of the software than the manuscript, but I wonder if the authors have plans to include import functions for data from specific devices, such as Omron (where a clumsy CSV export is available).

6. PLOS authors have the option to publish the peer review history of their article (what does this mean?). If published, this will include your full peer review and any attached files.

Reviewer #1: No

Reviewer #2: **Yes: **Tobias Jakobi

---

## [Author Response · Author response to Decision Letter 0]

15 Aug 2022

We thank the reviewers for their comments, and we have included our point by point response as a separate pdf compiled together with other submission files (the response is compiled in the end). The main changes are the following:

- Expanded justification of the benefits of R software platform over existing alternatives

- Significantly expanded discussion of limitations of the work and directions in need of further research

- Expanded software functionality to allow both default and customized adjustment of BP treshold values depending on BP type (HBPM or ABPM) with corresponding summary in the manuscript of the differences between office and out-of-office BP thresholds

- Expanded comments on and labelling of all code blocks to enhance readability and accessibility of examples

---

## [Editor Report · Decision Letter 1]

22 Aug 2022

bp: Blood Pressure Analysis in R

PONE-D-22-13672R1

Dear Dr. Gaynanova,

We’re pleased to inform you that your manuscript has been judged scientifically suitable for publication and will be formally accepted for publication once it meets all outstanding technical requirements.

Kind regards,

Fabiana Zama

Academic Editor

PLOS ONE

---

## [Editor Report · Acceptance letter]

29 Aug 2022

PONE-D-22-13672R1

bp: Blood Pressure Analysis in R

Dear Dr. Gaynanova:

I'm pleased to inform you that your manuscript has been deemed suitable for publication in PLOS ONE. Congratulations! Your manuscript is now with our production department.

Kind regards,

on behalf of

Professor Fabiana Zama

Academic Editor

PLOS ONE